Identifying CD1c as a potential biomarker by the comprehensive exploration of tumor mutational burden and immune infiltration in diffuse large B cell lymphoma

Xiang Xiaoyu
Gao Li-Min
Zhang Yuehua
Zhu Qiqi
Zhao Sha
Liu Weiping
Ye Yunxia
Tang Yuan 1202ty@163.com
Zhang Wenyan zhangwenyan@wchscu.cn
Department of Pathology, West China Hospital of Sichuan University , Chengdu, Sichuan , China
Amdare Nitin
Electronic publication date: 2023 Dec 11
Publication date: 2023
Volume: 11
Electronic Location ID: e16618
Received 2023 Mar 3; Accepted 2023 Nov 16
Copyright: © 2023 Xiang et al.
Copyright year: 2023
Copyright holder: Xiang et al.
License: This is an open access article distributed under the terms of the Creative Commons Attribution License, which permits unrestricted use, distribution, reproduction and adaptation in any medium and for any purpose provided that it is properly attributed. For attribution, the original author(s), title, publication source (PeerJ) and either DOI or URL of the article must be cited.
License URL: https://creativecommons.org/licenses/by/4.0/

Keywords: Diffuse large B cell lymphoma, Tumor mutational burden, Immune infiltration, CD1c, Prognostic biomarker, Immunotherapy

Funding: 1·3·5 Projects for Disciplines of Excellence–Clinical Research Incubation Project, West China Hospital, Sichuan University 2019HXFH035 This research was funded by 1·3·5 projects for disciplines of excellence–Clinical Research Incubation Project, West China Hospital, Sichuan University (No: 2019HXFH035). The funders had no role in study design, data collection and analysis, decision to publish, or preparation of the manuscript.

==============================
Background

Tumor mutational burden (TMB) is a valuable prognostic biomarker. This study explored the predictive value of TMB and the potential association between TMB and immune infiltration in diffuse large B-cell lymphoma (DLBCL).

Methods

We downloaded the gene expression profile, somatic mutation, and clinical data of DLBCL patients from The Cancer Genome Atlas (TCGA) database. We classified the samples into high-and low-TMB groups to identify differentially expressed genes (DEGs). Functional enrichment analyses were performed to determine the biological functions of the DEGs. We utilized the cell-type identification by estimating relative subsets of RNA transcripts (CIBERSORT) algorithm to estimate the abundance of 22 immune cells, and the significant difference was determined by the Wilcoxon rank-sum test between the high- and low-TMB group. Hub gene had been screened as the prognostic TMB-related immune biomarker by the combination of the Immunology Database and Analysis Portal (ImmPort) database and the univariate Cox analysis from the Gene Expression Omnibus (GEO) database including six DLBCL datasets. Various database applications such as Tumor Immune Estimation Resource (TIMER), CellMiner, konckTF, and Genotype-Tissue Expression (GTEx) verified the functions of the target gene. Wet assay confirmed the target gene expression at RNA and protein levels in DLBCL tissue and cell samples.

Results

Single nucleotide polymorphism (SNP) occurred more frequently than insertion and deletion, and C > T was the most common single nucleotide variant (SNV) in DLBCL. Survival analysis showed that the high-TMB group conferred poor survival outcomes. A total of 62 DEGs were obtained, and 13 TMB-related immune genes were identified. Univariate Cox analysis results illustrated that CD1c mutation was associated with lower TMB and manifested a satisfactory clinical prognosis by analysis of large samples from the GEO database. In addition, infiltration levels of immune cells in the high-TMB group were lower. Using the TIMER database, we systematically analyzed that the expression of CD1c was positively correlated with B cells, neutrophils, and dendritic cells and negatively correlated with CD8+ T cells, CD4+ T cells, and macrophages. Drug sensitivity showed a significant positive correlation between CD1c expression level and clinical drug sensitivity from the CellMiner database. CREB1, AHR, and TOX were used to comprehensively explore the regulation of CD1c-related transcription factors and signaling pathways by the KnockTF database. We searched the GETx database to compare the mRNA expression levels of CD1c between DLBCL and normal tissues, and the results suggested a significant difference between them. Moreover, wet experiments were conducted to verify the high expression of CD1c in DLBCL at the RNA and protein levels.

Conclusions

Higher TMB correlated with poor survival outcomes and inhibited the immune infiltrates in DLBCL. Our results suggest that CD1c is a TMB-related prognostic biomarker.

Introduction

Diffuse large B-cell lymphoma (DLBCL) is the most common type of aggressive non-Hodgkin lymphoma with a high morbidity and mortality rate (Liu & Barta, 2019). The standard treatment approach for patients with DLBCL is immunochemotherapy utilizing combinations of rituximab and anthracycline-based chemotherapy (Spinner & Advani, 2022). However, 30–40% of patients experience treatment failures or relapses, necessitating alternative treatment options (Poletto et al., 2022). In the past decade, targeted therapy and immunotherapy have emerged as promising alternatives for the management of DLBCL (Cheson, Nowakowski & Salles, 2021). Recent advances in understanding the molecular and immunological heterogeneity of DLBCL have led to the identification of novel therapeutic targets. Studies have reported frequent genetic alterations in various genes in DLBCL, including BCL2, BCL6, and MYC, which could be helpful targets for precision medicine approaches (Sermer et al., 2020).

Additionally, recognizing immune evasion mechanisms employed by tumor cells and developing immunotherapy agents offer new opportunities for durable and targeted treatment options for patients with DLBCL (Ye et al., 2022). Despite these advancements, DLBCL remains a challenging disease with a wide range of clinical subtypes, molecular alterations, and varied response rates to different treatments (Liu & Barta, 2019). Further research is needed to develop novel therapeutic approaches and better understand the complex interactions between the tumor microenvironment and DLBCL.

Recent advances in sequencing technologies have enabled the identification of genetic changes and mutations that play a crucial role in the pathogenesis and heterogeneity of DLBCL. Tumor mutational burden (TMB) is defined as the total number of mutations per megabase (Mb) of the genome. It has been shown to correlate with the efficacy of immunotherapy in various tumor types (Jardim et al., 2021). High TMB is associated with increased neoantigen production, which can promote a more robust immune response against the tumor. Therefore, TMB has garnered significant attention as a predictor of response rate to immune checkpoint inhibitors (ICIs) (Killock, 2020). However, the significance of TMB in DLBCL remains controversial. Recent studies have reported higher TMB in the ABC-DLBCL subtype than in the GCB-DLBCL subtype, indicating the potential of TMB as a prognostic marker and therapeutic target in ABC-DLBCL (Schmitz et al., 2018). A study tested 101 DLBCL patients and found that those with a higher TMB had poor OS and progression-free survival. Furthermore, specific genetic mutations such as MYD88, CD79B, and PIM1 were also associated with higher TMB levels, which suggests that TMB is a potential predictor of survival and prognosis in DLBCL patients (Chen et al., 2021).

Recent studies have shown the tumor microenvironment (TME) plays a crucial role in the pathogenesis and progression of DLBCL, and molecular targets related to the tumor microenvironment invasion may become the key to immunotherapy (Cioroianu et al., 2019). Based on advances in gene sequencing and expression profiling, studies have shown that the prognosis of DLBCL patients is related to TME (Ciavarella et al., 2019). For example, immune checkpoint inhibitors (ICI) have demonstrated significant efficacy in some refractory hematological malignancies (Thanarajasingam, Thanarajasingam & Ansell, 2016). These findings highlight the importance of studying the TME in DLBCL and suggest potential novel therapeutic strategies targeting the TME. Further research is needed to elucidate the complex interactions between diverse TME components and to identify biomarkers that can predict patient outcomes and guide personalized therapy. Tumor cells may evade immune recognition by the immune system by altering antigen presentation or modulating the TME, resulting in immune escape (Riaz et al., 2016). Therefore, exploring the relationship between TMB and immune regulation and identifying relevant genes or biological mechanisms will lead to a better understanding of the role of immunotherapy in cancer treatment.

In this study, we collected somatic mutation data, transcriptome data, and clinical information from the TCGA database, aiming to study the association between TMB and gene mutation, immune response, and prognosis of DLBCL combined with immune infiltration. We attempted to elucidate the relationships between TMB groups and clinicopathological factors, between TMB groups and different immune-infiltrating cells, and between TMB groups and prognosis. The results of these studies may provide novel biomarkers and potential treatment options for DLBCL.

Materials and Methods

Acquisition of somatic mutation data and expression profile from TCGA

We obtained the somatic mutation profile from the publicly available TCGA database via the GDC data portal (https://portal.gdc.cancer.gov/). Among the four subtypes of files, “Masked Somatic Mutation” data were selected and processed based on VarScan software. We were summarizing, analyzing, annotating, and visualizing mutation annotation format (MAF) files used to store detected somatic variation using the “maftools” Bioconductor package. In addition, we also downloaded the HTseq-FPKM transcription profile from UCSC Xena (https://xena.ucsc.edu/), which were respectively of TCGA-DLBC tumor samples and standard samples of “Cells-EBV-Transformed Lymphocytes” located in GETx database (https://www.gtexportal.org/home/). Corresponding clinical information was also collected from UCSC Xena, including age, sex, tumor grade, pathological stage, TNM stage, survival time, and OScensor.

Calculation of the TMB score of each sample and prognostic analysis

TMB values for each sample were determined by measuring the total number of nonsynonymous detected per million bases, which could be calculated as (whole counts of gene variants)/(the whole length of exons). In our study, we calculated the mutation frequency of variation/exon length (38 million) per sample based on the “maftools” R package. According to the median data, TCGA-DLBC samples were subdivided into low-(18 cases) and high-(19 cases) TMB groups. Then, TMB mutation data were combined with the corresponding survival data by sample ID, and Kaplan-Meier (KM) analysis was performed to compare survival differences between the low- and high-TMB groups with a log-rank sum test. In addition, the association between TMB and clinical features was further assessed, in which the Wilcoxon rank-sum test was used to calculate the P-value for the two groups, and the Kruskal-Wallis (KW) test was used for three or more groups.

Differentially expressed genes and functional pathways analysis

According to TMB level, the transcriptome profile was assigned into low- and high-TMB groups by R software. “limma” R package was used to identify the DEGs between the low- and high-TMB groups, and the thresholds were set at P < 0.05 and | log2 (fold change) | > 1.0. A heatmap was drawn by using the “pheatmap” R package. Then, the entreID of each DEG was generated using the “org.Hs.eg.db” R package, and we used the “clusterProfiler”, “GOplot”, and “ggplot2” R packages for Gene Ontology (GO) analysis and “enrichplot” for Kyoto Encyclopedia of Genes and Genomes (KEGG) enrichment analysis. Besides, we performed gene set enrichment analysis (GSEA; https://www.gsea-msigdb.org/gsea/index.jsp) (Hung et al., 2012) based on the JAVA8 platform using the TMB group as the phenotype and TCGA-DLBC mRNA expression profile as expression spectrum data file. Then we selected the “c2.cp.kegg.v6.2.symbol.gmt” gene set as a reference gene set, which is derived from the MsigDB Database (https://www.gsea-msigdb.org/gsea/msigdb). The significant enrichment pathway was considered only when P < 0.05.

Co-analyses of TMB and immune infiltration

We evaluated the proportion of immune cells using the deconvolution algorithm CIBERSORT (https://cibersort.stanford.edu/). CIBERSORT (CIBERSORT R Script V1.03) was a general calculation method that can accurately estimate the composition of 22 immune cells in tumor tissues by combining the prior knowledge of the composition spectrum of purified leukocyte subsets with support vector regression (Newman et al., 2015). We then identified the differences in the composition of immune cells between low-and high-TMB groups, and the number of permutations was set to 1,000 as well as P < 0.05. The “pheatmap” R package showed the distribution of immune cells between the two groups, and the “vioplot” package was used to indicate the differential immune infiltration by the Wilcoxon rank-sum test. The threshold P < 0.05 was the standard to calculate the significance of a single immune cell between the two groups. In addition, we obtained a list of 2,483 immune-related genes from the Immunology Database and Analysis Portal (https://www.immport.org/). The “VennDiagram” R package was used to screen the intersecting genes between TMB-DEGs and immune-related genes. Univariate Cox regression analysis was performed to determine the predictive TMB-related immune genes using “survival”, “survminer”, and “forestplot” R packages.

Validating the prognostic TMB-related immune genes in the GEO database

We systematically searched for the GEO database (https://www.ncbi.nlm.nih.gov/geo/) open clinical annotations of DLBCL gene expression profile data and obtained the three datasets, including GSE10846, GSE31312, GSE32918, GSE53786, GSE87371, GSE181063. Then proceed with dataset processing, (i) Data downloading, downloading dataset file of a series matrix; (ii) Background correction and standardization of data, such as quantile standardization; (iii) Using GPL570 and GPL8432 annotation files for ID translation; (iv) The same gene corresponds to multiple probes, and the average value of the probes was calculated as the expression level; (V)Complete expression profile data files and corresponding clinical information of patients, including survival time and survival status, were obtained. As a result, 1133 DLBCL samples were selected, including 414 (GSE10846), 470 (GSE31312), 249 (GSE32918), GSE53786 (N = 119), GSE87371 (N = 221), GSE181063 (N = 1,149) samples. The Combat function in the “sva” R package removed the batch effect and integrated three datasets to obtain the expression spectrum. Prognostic TMB-related immune genes were screened to verify whether there was a statistical significance between their expression and prognosis in GSE datasets. We selected five TMB-related immune genes with |log2 FC| > 1 and P < 0.05 to further assess the predictive value of differential immune genes in patients with low-and high-TMB levels. Kaplan-Meier analysis was conducted via a “for cycle” R script to find the hub immune genes associated with survival outcomes.

Copy number variations (CNV) and correlated immune cells of the prognostic TMB-related immune genes

The Tumor Immune Estimation Resource database (TIMER, https://cistrome.shinyapps.io/timer/), a web server for comprehensive analysis of tumor-infiltrating immune cells, was used to estimate the abundance of six types of immune infiltrating cells, such as B cells, CD4+ T cells, neutrophils, macrophages, and dendritic cells (Li et al., 2020). Changes in copy Number Variations (CNV) were observed in prognostic TMB-related immune genes, and the correlations between CNV and immune cells abundance and between immune cells and survival were further assessed.

Analysis of drug sensitivity and transcription factors signaling pathway of the target genes

The drug sensitivity data used in this study were obtained from the CellMiner database (https://discover.nci.nih.gov/cellminer/home.do) (Reinhold et al., 2012). The transcriptome and drug sensitivity data of the same batch of samples were downloaded, and the expression profile of the target gene and drugs verified by the Food and Drug Administration (FDA) were retained by sorting the data. Then, the correlation between target gene expression level and drug sensitivity was extracted and further explored by Spearman correlation analysis. The higher the correlation value, the stronger the correlation. KnockTF (http://www.licpathway.net/KnockTF/index.html) was used to examine a combination of the regulation of gene-related transcription factors and log|FC| > 1.0 signaling pathways (Feng et al., 2020).

Pathological specimens and clinical parameters of the patients

We searched the GETx database (https://www.gtexportal.org/home/) to compare mRNA expression levels of the target genes between lymphoma tissues and normal tissues. The difference in CD1c mRNA expression between tumor and normal tissues was verified by wet assay. One hundred ten paraffin samples from 2021.12 to 2022.2 from the Department of Pathology of West China Hospital of Sichuan University were screened, of which 76 cases were confirmed as DLBCL samples and 34 samples of normal lymphoid tissue hyperplasia. The enrollment requirements consisted of the following: (1) Patients who received a biopsy at the Department of Pathology of West China Hospital, Sichuan University, and were diagnosed with primary diffuse large B-cell lymphoma, with comprehensive pathological records; (2) patients with complete clinical data and follow-up information. The exclusion criteria were Patients with insufficient pathological diagnostic data or incomplete clinical information. This study (IRB: 2020-703) was approved by the Biomedical Research Ethics Committee of West China Hospital, Sichuan University, and informed consent was waived.

Immunohistochemistry (IHC) assays

All tissues were embedded in paraffin and cut into 4 µm sections. Sections were dewaxed with xylene and ethanol. Sections were then repaired in EDTA antigen repair solution (PH = 9) at high temperature and cooled to 25 °C. 3% H2O2 was used to seal endogenous peroxidase. Sections were then incubated with CD1c primary antibody (1:50; Cat# ab156708, Abcam Inc., Cambridge, UK) at 4 °C overnight. The secondary antibody (Cat# GK500705, Dako Agilent Inc., Glostrup, Denmark) was incubated for 2 h. Diaminobenzidine (DAB) was used for the color development reaction. Hematoxylin staining was performed for 20 s, dehydrated, transparent and sealed. All images were captured by a digital pathology slide scanner (KF-PRO-005-EX; KFBIO Technology for Health Co. Ltd.).

Grading criteria

We evaluated the staining reaction based on the immunoreactive score (IRS) created by Remmele and Stegner. IRS = SI (staining intensity) × PP (percentage of positive cells). CD1c immunostaining was positive for the cell membrane. Positive cells appeared yellow to brown. The positive cells were defined as clearly located positive cells observed under a light and high magnification microscope. All stained slides were independently evaluated by two pathologists. The immunohistochemical staining slides were scanned by KFBIO Digital Pathology Slide Scanner, and the scanning results were analyzed by K-Viewer software. Three ROI (region of interest) regions were selected for each section for analysis. Based on the intensity of positive staining, the sections were graded as follows: 0 points, negative; one point, weak; two points, moderate; and three points, strong. The H-Score (0–300) was calculated by multiplying the staining intensity score (0–3) by the percentage of positive cells covering the area (0–100) of each positive intensity (1+, 2+, 3+) in a given area.

Cell culture

SU-DHL-2 (Activated B-cell, ABC subtype), SU-DHL4 (Germinal center B-cell, GCB subtype), and HMy.C1R (normal human B lymphoblastic-cell) cell lines were purchased from the Chinese Academy of Sciences (Shanghai, China) and cultured in RPMI-1640 medium (Gibco; Thermo Fisher Scientific, Inc., Waltham, MA, USA) and IMDM medium (Gibco; Thermo Fisher Scientific, Inc., Waltham, MA, USA) supplemented with 10% fetal bovine serum (Gibco; Thermo Fisher Scientific, Inc., Waltham, MA, USA), 100 units/ml penicillin and 100 μg/ml streptomycin (Thermo Fisher Scientific, Inc., Waltham, MA, USA).

Western blotting

Protein was extracted from RIPA lysate (Shanghai Epizyme Biomedical Technology Co., Ltd, Shanghai, China) and separated by 10%SDS-PAGE gel (Germany biofroxx Biomedical Technology Co., Ltd). The protein was then transferred to a PVDF membrane (Biosharp Biotechnology), blocked with 5%BSA for 2 h, and bound to a primary antibody, including CD1c (1:1,000; Cat#ab156708; Abcam Inc., Cambridge, UK), β-tubulin (1:10,000; Cat# ET1602-4; Hangzhou Huaan Biotechnology Co., Ltd, Hangzhou, China). After washing the membrane, we incubated it with the secondary antibodies (1:5,000; Cat# LK2003; SUNGENE Biotech, Bengaluru, Karnataka). Protein images were captured using a chemiluminescence system (BioRad Laboratories, Hercules, CA, USA).

Quantitative reverse transcription polymerase chain reaction (qRT-PCR)

Total RNA was extracted from FFPE samples, and gDNA was removed using the RNApure FFPE kit (CW0535; CoWin Bioscience, Beijing, China), following the manufacturer’s protocol. HiScript® III All-in-one RT SuperMix (R333; Vazyme, Nanjing, China) was utilized for reverse transcription, and the synthesized cDNA was employed as a template for real-time fluorescence quantification. Real-time quantitative PCR (RT-qPCR) was conducted on a Real-time PCR Detection System (Bio-rad) using SYBR® Green Premix Ex Taq™ II (Tli RNaseH Plus) (RR820A; TaKaRa, Beijing, China). Independent experiments were performed in triplicate, with ß-actin as an internal control. The following primers (Tsingke Biotechnology Co., Ltd., Beijing, China) were used: CD1c: FP 5′-CACTTGCCCCCGATTTCTCT-3′; RP 5′-ATGGAAAAGTGGTGTCCCCAG-3′. ACTIN: FP 5′-CCGCGAGAAGATGACCCAGA-3′; RP 5′-GATAGCACAGCCTGGATAGCA-3′.

Statistical analysis

Student’s t-test was used for continuous variables, while categorical variables were compared by χ2 test. The Wilcoxon rank-sum test was a nonparametric statistical test mainly utilized for comparing two groups. The Kruskal-Wallis test was suitable for two or more groups. Fisher’s exact test, or the χ2 test, was used to analyze clinical characteristic data. Overall survival (OS) was analyzed by Kaplan-Meier curves and log-rank tests. Immune cells were analyzed by Spearman rank correlation. All statistical analysis was achieved in R (Version 4.2.3) and GraphPad Prism 8 software. We considered *P < 0.05, **P < 0.01, ***P < 0.001 and #P < 0.0001 were considered statistically significant.

Results

The landscape of mutation profiles in DLBCL

The research strategy is presented in (Fig. 1). Somatic mutation profiles of 37 DLBCL samples were downloaded from the TCGA database. We used the “maftools” R package to visualize mutation data in VAF format. In general, missense mutation accounted for the most significant proportion of mutation types (Fig. 2A), and the occurrence frequency of single nucleotide polymorphism (SNP) was higher than insertion and deletion (Fig. 2B). The most common type of base substitution was C>T (Fig. 2C). The boxplot (Figs. 2D and 2E) showed different mutation types in DLBCL patients, and Fig. 2F showed the top 10 genes with mutation frequency, including PIM1 (22%), IGLV3-1 (38%), IGLL5 (27%), IGHG1 (22%), IGHV2-70 (27%), BTG (27%), IGHM (24%), KMT2D (32%), IGLC2 (24%), CARD11 (22%). The mutation landscape displayed the mutation information of each sample, in which the mutation frequency of IGLV3-1 and KMT2D accounted for 38% and 32%, respectively (Fig. 2G). Heatmap of gene correlations shows gene-to-gene relationships. For example, there is a synergistic effect between MUC16 and FAT4, while SOCS1 and KMT2D are mutually exclusive (Fig. 2H). Meanwhile, the Genecloud plot displayed the frequency of mutations in genes (Fig. S1), and the higher the mutation frequency, the larger the gene name.

Figure 1 The workflow of the study.

Figure 2 Summary of the mutation information with statistical.

(A–C) Classification of mutation types according to different categories, in which missense mutation accounts for the most fraction. SNP showed more frequency than insertion or deletion, and C > T is the most common Single nucleotide variant (SNV). (D and E) TMB in specific samples. (F) The top 10 mutated genes in DLBCL. (G) The landscape of mutation profiles in DLBCL samples. Mutation information of each gene in each sample is shown in the waterfall plot, in which various colors with annotations at the bottom represent the different mutation types. The barplot above the legend exhibits the mutation burden. (H) The coincident and exclusive associations across mutated genes. (DLBCL, diffuse large B cell lymphoma; TMB, tumor mutational burden; SNP, single nucleotide polymorphism; SNV, single nucleotide variants).

TMB correlated with survival outcomes and clinical pathological characteristics

We calculated the mutation event per million bases as the TMB for DLBCL patients, worked out the optimal cutpoint using the surv_cutpoint function in the “survival” R package, and set the parameter cutpoint = 2.8 to divide patients into low- (18 cases) and high-(19 cases) TMB groups. TMB ranged from 0.14 to 6.92 with a median of 1.9 per MB (Fig. 3A). Kaplan-Meier survival analysis was carried out, and the result showed that the 5-year survival rate of the high-TMB group was lower than the low-TMB group (Fig. 3B). In addition, none of the clinical traits was significantly correlated with TMB level, which may be due to the small samples (Table S1).

Figure 3 Distribution of TMB samples and prognosis of TMB.

(A) Distribution of TMB samples: those above the median value represent the samples with high mutation, and those below the median value represent the samples with low mutation. (B) Higher TMB levels are associated with poor survival outcomes with a P-value = 0.076. (TMB, tumor mutational burden).

Identifying differentially expressed genes based on TMB grouping and functional enrichment analysis of GO, KEGG, and GSEA

Differentially expressed genes (DEGs) were calculated by R software version 4.2.2 (R Core Team, 2023). A total of 62 DEGs were identified in the low- and high-TMB group using the “limma” R package by setting the P < 0.05 and | log2 (FC) | > 1 (Table 1). The heatmap visualized DEGs between the low- and high-TMB groups (Fig. 4A). The volcano map showed 42 up-regulated genes and 20 down-regulated genes (Fig. 4B). Subsequently, we conducted GO enrichment analysis on DEGs and found that the differential genes were mainly involved in immune-related pathways, such as lymphocyte-mediated immunity, adaptive immune response based on somatic recombination of immune receptors built from immunoglobulin superfamily domains, immunoglobulin mediated immune response and complement activation, classical pathway (Fig. 4C and Table S2). The enrichment information of the GO pathways is illustrated in Fig. 4D. CD1c, CCL21, TP63, ORM1, ACTG2, IGHG3, IGHM, TRPM4, and so on are involved in all of the top GO pathways (including Molecular Function, Cellular Component, Biological Process) and were identified as hub genes. KEGG pathways illustrated that the differential genes were mainly enriched in vascular smooth muscle contraction, hematopoietic cell lineage, carbon metabolism, and neuroactive 210 ligand−receptor interaction pathway (Fig. S2 and Table S3). In addition, we further selected the GSEA results of the top TMB-related items in Figs. 4E–4G, including one carbon pool by folate, rig-i like receptor signaling pathway-creative diagnostics, and the tight junction, which were associated with the TMB level (P < 0.05).

Table 1 Differentially expressed genes between low-TMB and high-TMB groups.

Gene symbol	logFC	AveExpr	t	P.Value	adj.P.val	B	
PRAME	2.922405	1.672344	5.237783	8.14E−06	0.184374	2.785421	
SLC12A3	1.100708	0.871668	5.168989	1.00E−05	0.184374	2.580099	
EIF5AP2	1.037704	0.7309014	4.540696	6.55E−05	0.446506	0.725776	
TP63	1.498609	1.72559	4.29477	0.000135	0.446506	0.015043	
C17orf99	1.701547	1.3456096	4.276447	0.000143	0.446506	−0.03746	
CNTNAP4	1.047761	0.3234867	4.189251	0.000184	0.446506	−0.28634	
LINC02562	1.440104	0.682375	3.902082	0.000421	0.597317	−1.09357	
LY86	−1.36579	4.6202479	−3.65037	0.000859	0.628234	−1.78272	
GTSF1	2.491803	3.3402724	3.647969	0.000865	0.628234	−1.78919	
TRPM4	2.112655	2.5636372	3.589926	0.001018	0.685802	−1.94525	
AL627309.6	1.130337	1.5450149	3.531998	0.001195	0.691416	−2.09986	
MIR5195	−1.90974	4.7542273	−3.45277	0.001487	0.77525	−2.30939	
TCTN1	1.056882	1.3681228	3.418234	0.001635	0.799445	−2.40003	
ASB2	−1.38053	3.2815049	−3.30631	0.002216	0.84459	−2.69058	
SLC9A7	1.001124	3.0139012	3.296243	0.002277	0.84459	−2.71648	
AL627309.7	1.147779	1.3959171	3.215333	0.002829	0.902441	−2.92307	
ORM1	2.689855	2.1889532	3.185368	0.003065	0.934341	−2.99888	
IL4I1	1.275177	5.8378104	3.159182	0.003286	0.945784	−3.06482	
PES1P2	1.059463	0.3808462	3.144121	0.003419	0.955589	−3.10261	
ACTG2	2.614232	2.8996668	3.131267	0.003538	0.96048	−3.13478	
PTGIR	1.373384	2.2671073	3.119858	0.003646	0.96048	−3.16327	
IGKV1D-8	−1.26761	1.4899645	−3.11549	0.003688	0.96048	−3.17417	
AP000593.3	1.96134	1.6328582	3.061174	0.004254	0.979536	−3.30891	
SSTR2	1.027005	1.0412898	3.029081	0.004626	0.999998	−3.38788	
AF127936.1	1.028164	0.7326819	2.986073	0.005172	0.999998	−3.49296	
ATF5	1.412084	6.6579457	2.967457	0.005427	0.999998	−3.53817	
IGHV5-78	−1.64874	4.4896813	−2.9616	0.00551	0.999998	−3.55236	
ARLNC1	1.842147	2.1616004	2.883652	0.006729	0.999998	−3.7396	
OTOF	1.054667	0.5352778	2.858965	0.007165	0.999998	−3.79827	
AL137026.2	1.234373	0.6788708	2.818765	0.007932	0.999998	−3.89314	
CTSLP2	1.010085	0.3901514	2.797078	0.008378	0.999998	−3.94397	
NFIL3	1.002554	3.2248748	2.796697	0.008386	0.999998	−3.94486	
AC005083.1	−1.00218	1.4038589	−2.70415	0.010562	0.999998	−4.15895	
MIR4538	−2.1038	2.6705054	−2.65241	0.011997	0.999998	−4.2766	
PSMA8	1.058776	0.8946082	2.611077	0.01327	0.999998	−4.3695	
ORM2	1.39914	1.4257549	2.597839	0.013704	0.999998	−4.39905	
DNAJC5B	1.282866	2.7580284	2.563015	0.014907	0.999998	−4.47631	
AC024475.4	1.115043	0.6922118	2.532595	0.016037	0.999998	−4.54321	
GPR160	−1.02655	2.3532824	−2.52493	0.016334	0.999998	−4.55998	
GLDC	−1.14561	1.4819458	−2.50791	0.017011	0.999998	−4.5971	
KRT17	1.215934	0.8071803	2.505569	0.017106	0.999998	−4.60218	
FAM129A	1.01797	2.6609447	2.484965	0.017965	0.999998	−4.64685	
LRRC32	−1.22325	3.030123	−2.48445	0.017987	0.999998	−4.64797	
IGHG3	−2.28464	5.678394	−2.48286	0.018055	0.999998	−4.65141	
ETV7	1.003919	2.1068744	2.45744	0.019174	0.999998	−4.70613	
CR2	−1.8187	3.1337495	−2.44458	0.019763	0.999998	−4.73365	
CD1C	−1.56179	2.9112972	−2.42331	0.020775	0.999998	−4.77898	
TREML2	1.251118	2.1785378	2.409244	0.02147	0.999998	−4.80879	
TREM1	1.013126	1.0170469	2.369286	0.02356	0.999998	−4.89281	
USP2	1.059806	0.9148026	2.362492	0.023933	0.999998	−4.907	
RPL10P6	1.261382	2.1649398	2.339933	0.025211	0.999998	−4.95389	
RPL10P9	1.236246	3.0773338	2.337468	0.025354	0.999998	−4.95899	
PPP1R14A	1.094098	1.1728365	2.324857	0.026099	0.999998	−4.98504	
CCL21	−2.82779	5.2870774	−2.27481	0.029253	0.999998	−5.08742	
FBP2	1.072932	0.6428013	2.256376	0.030498	0.999998	−5.1247	
TRBJ2-2P	−1.42691	3.7776966	−2.24323	0.031415	0.999998	−5.15115	
TRBJ2-1	−1.05611	2.6341454	−2.16798	0.037154	0.999998	−5.30037	
IGKV1-5	−1.88826	3.856146	−2.16686	0.037245	0.999998	−5.30255	
CPA6	1.006875	0.7805608	2.165899	0.037325	0.999998	−5.30444	
IGHV1-69	−1.05659	1.4366253	−2.14157	0.039378	0.999998	−5.35181	
IGHV3-23	−1.56581	3.7508495	−2.08555	0.044491	0.999998	−5.45934	
IGHM	−2.13098	9.540321	−2.04243	0.048817	0.999998	−5.54061	

Figure 4 Comparisons of gene expression profiles in low- and high-TMB groups and enrichment pathway analysis.

(A) A total of 62 DEGs are shown in the heatmap plot. Vertical and horizontal axes represent genes and DLBCL samples, respectively. Gene expression levels with higher and lower were displayed in red and blue, respectively. Color bars at the top of the heatmap represent sample types, with red and green indicating low- and high-TMB samples, respectively. (B) Volcano plots of all DEGs were drawn with | log2 (FC) | > 1 and p-value < 0.05. Each symbol represents a gene, and red, grey, and blue indicate upregulated, normal, and downregulated genes, respectively. (C) GOplot reveals that these differentially expressed genes are involved in immune-related pathways. Different colors represent different GO terms, and the depth of gene color means log2 (FC). (D) The DEGs enrichment analysis information (red represents the pathway for CD1c gene enrichment). (E and F) GSEA analysis showed high-TMB-related crosstalks, including one carbon pool by folate and rig-i-like receptor signaling pathway-creative diagnostics. (G) GSEA analysis shows that low-TMB-related crosstalk, including tight junction. (DLBCL, diffuse large B cell lymphoma; TMB, Tumor mutational burden; NES represents the normalized enrichment score; ES represents enrichment score; DEGs, differentially expressed genes; GO, gene ontology; GSEA, gene set enrichment analysis; FC, fold change).

Differential abundance of immune cells in the low- and high-TMB groups using CIBERSORT

After DEG screening, to further compare the difference in the degree of immune cell infiltration between low- and high-TMB groups, we calculated the composition ratio of immune cells per sample by the “CIBERSORT” R package. The boxplot in Fig. 5A showed a specific portion of 22 immune cells in each DLBCL sample. We also calculated the proportion of immune cells in the whole DLBCL cohort, accounting for the most, including B cells naive, CD8+ T cells, M2 macrophages, and M0 macrophages (Fig. 5B). The heatmap showed the distribution of immune cells between low- and high-TMB groups, and the result displayed that the high-TMB group had a lower immune score (Fig. 5C). In addition, the Wilcoxon rank-sum test demonstrated that monocytes, dendritic cells activated, and dendritic cells resting were more deficient in the high-TMB group (P < 0.05) (Fig. 5D). According to the above analysis results, the high-TMB group inhibited the level of immune cells infiltration in DLBCL samples.

Figure 5 Relationship between TMB and immune infiltration.

(A) The stacked bar chart shows the distribution of 22 types of immune cells in each sample. The horizontal axis represents the sample name, and the vertical axis represents the proportion of 22 types of immune cells. (B) The boxplot is arranged according to the content of immune cells in all DLBCL samples, among which B cells naïve accounted for the most significant proportion. (C) The difference analysis of the heatmap shows the distribution of immune cells in the low- and high-TMB samples. (D) The boxplot indicates differentially infiltrated immune cells between low- and high-TMB groups, with green representing the high-TMB group and red representing the low-TMB group. (DLBCL, diffuse large B cell lymphoma; TMB, Tumor mutational burden).

Screening TMB-related immune genes and verifying the prognosis of the screened genes using the GEO database

The immune-related genes were downloaded from the ImmPort database and intersected with the selected DEGs. A total of 13 TMB-related immune genes were obtained, including CD1c, ORM1, ORM2, CCL21, CR2, IGHG3, IGHM, IGHV1-69, IGHV3-23, IGKV1-5, IGKV1D-8, SSTR2, TRBJ2-1 (Fig. 6A). Subsequently, univariate regression analysis was performed on the above genes, and it was found that there was no significant correlation between these genes and prognosis (P > 0.05, Fig. S3), possibly due to the small samples in the TCGA-DLBC cohort. Therefore, we expanded the sample size and screened a total of 1,133 samples of DLBCL gene expression microarray datasets (GSE31312, GSE10846, GSE32918) from the GEO database, as well as the clinical information of the corresponding samples. After ID translation, data homogenization and standardization, and removal of batch effect, five genes including CD1c, CCL21, ORM1, CR2, and SSTR2 (the rest of the eight genes were not included in the expression profile data) were obtained after joint analysis of the three datasets. Kaplan-Meier survival analysis was performed, and the results showed that the level of CD1c expression was significantly correlated with prognosis (P < 0.05, Figs. 6B–6F). Kaplan Meier survival curves were plotted in six GEO datasets, including GSE31312, GSE10846, GSE32918, GSE53786, GSE87371, and GSE181063, respectively, and the results showed that the prognosis was poor in the high CD1c expression groups (Fig. S4).

Figure 6 Identification of important TMB-related immune genes for DLBCL prognosis.

(A) Venn diagram shows that 13 differential immune genes are associated with TMB and immune infiltration; Kaplan-Meier survival analysis shows a relationship between the expression of CD1c, CCL21, ORM1, CR2, and SSTR2 and the prognosis, suggesting that down-regulation of CD1c is associated with better survival outcomes. (B) CD1c (P-value = 0.0012). (C) CCL21 (p-value = 0.0862). (D) ORM1 (P-value = 0.1745). (E) CR2 (P-value = 0.3151). (F) SSTR2 (P-value = 0.0715). (DLBCL, diffuse large B cell lymphoma; TMB, tumor mutational burden).

CNV of CD1c, immune cells, and survival in DLBCL using TIMER database

In general, CNV refers to an increase or decrease in the copy number of large segments of the genome that are more than 1 kb in length. CNV was observed in prognostic TMB-related immune genes, and to verify the CNV of CD1c and the relationship between immune cell content and prognosis, we utilized the TIMER database to obtain CD1c expression between normal and tumor tissues in various cancers, and it shows that the expression level of CD1c is the highest in Thymoma (THYM), followed by that of DLBCL (Fig. 7A). Especially in DLBCL, CD1c expression was positively correlated with B cells, neutrophils, dendritic cells and negatively associated with CD8+ T cells, CD4+ T cells, and macrophages, among which, the correlation with B cells was the highest (cor = 0.693, P = 1.44E−03, Fig. 7B). In addition, high amplification of CD1c was significantly different compared to other CNVs (P < 0.01, Fig. 7C). As for the relationship between immune cells content and prognosis, high levels of CD8+ T cells and dendritic cells indicate a superiors survival result. In contrast, low expression of CD1c may promote better survival (Fig. 7D).

Figure 7 Correlations between the CNV of CD1c, immune cell infiltration, and prognosis using the TIMER database.

(A) CD1c mRNA expression between normal and tumor tissues in various cancers, the horizontal axis represents the tumor types in the TCGA database, and the vertical axis represents the CD1c mRNA expression level (log2 TPM). (B) The expression of CD1c is correlated with six types of immune infiltrating cells, of which the correlation with B cells was the highest (cor = 0.693, P-value = 1.44e−03). The horizontal axis represents the immune cells infiltration level, and the vertical axis represents the CD1c expression level (log2 TPM). (C) High amplification of CD1c in B cells and dendritic cells (P-value < 0.01), the horizontal axis represents six types of immune cells from TIMER data, and the vertical axis represents the immune cells infiltration level. (D) High levels of CD8+ T cells and dendritic cells indicate a good prognosis; the horizontal axis represents survival time (months), and the vertical axis represents survival probability. (CNV, Copy number variation; TIMER, tumor immune estimation resource; TPM, transcripts per million; cor, correlation; *P < 0.05, **P < 0.01, ***P < 0.001).

Relationship between CD1c and drug sensitivity and regulation of CD1c transcription factors

According to the correlation analysis of target genes and drug sensitivity in the CellMiner database, it was found that there was a significant correlation between the expression level of CD1c and clinical drug sensitivity, mainly with nelarabine, methylprednisolone, chelerythrine, ribavirin, fluphenazine was positive (Figs. 8A and 8B). Therefore, the higher the expression of CD1c, the more sensitive cells are to these drugs. The KonckTF database results showed that the regulation of CD1c may be related to the effects of transcription factors such as CREB1, AHR, and TOX, resulting in the corresponding biological effects (Table 2).

Figure 8 Analysis of the relationship between TMB target gene and drug sensitivity.

(A) The scatter plot based on | cor | value of the top 20 targeted small molecule drugs sensitivity and the expression of CD1c, the horizontal axis for CD1c expression, and the vertical axis of IC50. (B) The lollipop plot also shows the relationship between CD1c expression and drug sensitivity, with the p-value indicating significance and cor indicating correlation, the horizontal axis representing the correlation, and the vertical axis representing the 20 targeted small-molecule drugs. The point size represents the absolute correlation value, and the color depth represents the P-value. (IC50, half maximal inhibitory concentration; TMB, tumor mutational burden; TPM, transcripts per million; cor, correlation).

Table 2 Transcription factors regulating CD1c in the Haematopoietic and lymphoid tissue by knockTF database.

Target gene	TF	Knock-method	Tissue type	Biosample name	Fold change	Log2FC	
CD1c	CREB1	shRNA	Haematopoietic and lymphoid tissue	K562	0.43886	−1.18818	
CD1c	AHR	siRNA	Haematopoietic and lymphoid tissue	THP-1	0.36694	−1.4464	
CD1c	TOX	shRNA	Haematopoietic and lymphoid tissue	CCRF-CEM	0.32445	−1.62392	

Verified the expression of CD1c at mRNA and protein levels

CD1c mRNA levels were compared between normal tissues downloaded by GETx and tumor tissues of TCGA-DLBC. We found a significant difference in the expression level of CD1c between normal and tumor tissue (P < 0.05, Fig. S5). We performed RT-qPCR to detect the expression of CD1c mRNA in three cell lines, and the results showed that SU-DHL-4 (P = 0.0017) had the highest CD1c expression level, followed by SU-DHL-2 (P = 0.0008) and HMy2.CIR had the lowest expression level (Fig. 9A). We collected 76 DLBCL pathological sections and 34 sections of reactive lymphoid hyperplasia (Table 3). CD1c was significantly overexpressed in DLBCL tissues compared with normal lymphoid tissue hyperplasia (P < 0.0001, Fig. 9B). Western blot results demonstrated that SU-DHL-4 cell lines expressed high levels of CD1c protein. In contrast, the HMy.C1R cell line showed low expression of CD1c protein. It should be noted that SU-DHL-2 may be due to the cell line activity being too low, with no reference gene β-Tubulin expression (Fig. 9C). The immunohistochemical results showed that in 50 DLBCL tissue samples, 20 samples had high CD1c expression, and 30 samples had low CD1c expression. The positive rate of CD1c in DLBCL reached 90%. Among 19 cases of reactive lymphoid hyperplasia, 12 had low CD1c expression, and seven had high CD1c expression. The positive rate in the control group was 68.42% (Table S4). The protein expression of CD1c in DLBCL tissues was significantly higher than in reactive lymphoid hyperplasia tissues (Figs. 9D and 9E). Wet experiments further confirmed the reliability of the bioinformatics findings. To investigate the relationship between long-term prognosis and CD1c, we obtained OS data according to CD1c expression in DLBCL patients. The results showed that the prognosis of the CD1c high-expression group was poor. However, the difference was not statistically significant for the high and low CD1c protein expression groups, possibly due to the small sample size (Figs. 9F and 9G).

Figure 9 The expression characteristics of CD1c in retrospectively collected DLBCL cell lines and clinical samples.

(A) CD1c mRNA expression in DLBCL and normal cell lines. (B) CD1c mRNA expression in tumor tissue (N = 76) and reactive lymphoid hyperplasia (N = 34). (C) CD1c protein expression in DLBCL and normal cell lines. (D) The representative samples reveal the high expression of CD1c in tumor tissue compared with Reactive lymphoid hyperplasia. (E) CD1c protein expression in DLBCL samples (N = 50) and reactive lymphoid hyperplasia (N = 19). (F and G) The Kaplan–Meier survival curve shows that high mRNA and protein expression of CD1c was related to poor overall survival in patients from the DLBCL clinical cohort. (*P < 0.05, **P < 0.01, ***P < 0.001, #P < 0.0001).

Table 3 76 DLBCL samples clinical information.

Sample	SampleID	PaththID	Gender	Age	Sites	Hans	Status	OStime	
Sample01	T1	L2000519	Male	69	Nodal	ABC	0	1,393	
Sample02	T2	L2000214	Female	64	Nodal	ABC	1	38	
Sample03	T3	L2100150	Male	47	Nodal	ABC	0	709	
Sample04	T4	L2100157	Female	56	Nodal	ABC	0	695	
Sample05	T5	L2100202	Female	80	Nodal	NA	0	653	
Sample06	T6	L2100204	Female	65	Nodal	GCB	0	674	
Sample07	T7	L2100213	Male	59	Nodal	ABC	0	658	
Sample08	T8	L2100246	Female	35	Nodal	ABC	0	624	
Sample09	T9	L2100254	Female	66	Nodal	ABC	0	660	
Sample10	T10	L2100271	Female	63	Nodal	ABC	0	1,089	
Sample11	T11	L2100287	Female	43	Nodal	ABC	0	581	
Sample12	T12	L2100408	Male	49	Nodal	ABC	1	544	
Sample13	T13	L2100295	Female	56	Nodal	ABC	0	620	
Sample14	T14	L2100417	Male	71	Nodal	ABC	1	4	
Sample15	T15	L2100415	Male	49	Nodal	ABC	0	513	
Sample16	T16	L2100454	Male	61	Nodal	ABC	0	498	
Sample17	T17	Z2139511	Female	59	Nodal	ABC	0	631	
Sample18	T18	Z2152204	Female	94	Nodal	ABC	1	98	
Sample19	T19	Z2152985	Male	48	Nodal	ABC	0	507	
Sample20	T20	Z2157414	Female	77	Nodal	ABC	0	504	
Sample21	T21	L1900028	Male	30	Nodal	ABC	0	1,514	
Sample22	T22	L1900160	Male	21	Nodal	ABC	0	1,431	
Sample23	T23	L1900164	Male	59	Nodal	ABC	1	1,153	
Sample24	T24	L1900191	Female	66	Nodal	ABC	0	1,428	
Sample25	T25	Z1919871	Male	72	Gastrointestinal	ABC	1	1,198	
Sample26	T26	Z1927492	Female	72	Gastrointestinal	ABC	0	1,380	
Sample27	T27	Z1959932	Male	38	Gastrointestinal	ABC	0	1,232	
Sample28	T28	Z1931327	Male	78	Gastrointestinal	ABC	1	421	
Sample29	T29	Z1924788	Male	44	Gastrointestinal	ABC	0	1,366	
Sample30	T30	Z1918661	Male	64	Nodal	GCB	0	1,400	
Sample31	T31	Z1905302	Male	84	Nodal	ABC	1	376	
Sample32	T32	Z1911133	Female	66	Gastrointestinal	ABC	1	1,258	
Sample33	T33	Z1941053	Female	34	Nodal	ABC	1	42	
Sample34	T34	Z1934519	Male	65	Gastrointestinal	GCB	0	1,348	
Sample35	T35	Z1933930	Male	55	Gastrointestinal	GCB	1	26	
Sample36	T36	Z1900756	Male	57	Gastrointestinal	ABC	1	1,409	
Sample37	T37	Z2008503	Male	79	Gastrointestinal	ABC	0	1,078	
Sample38	T38	Z2013238	Male	49	Gastrointestinal	GCB	0	1,049	
Sample39	T39	Z2016528	Male	70	Nodal	GCB	1	1,209	
Sample40	T40	Z2017865	Female	77	Gastrointestinal	ABC	0	1,025	
Sample41	T41	Z2017001	Male	51	Gastrointestinal	ABC	1	487	
Sample42	T42	Z2043475	Female	53	Gastrointestinal	ABC	1	290	
Sample43	T43	Z2121652	Male	31	Gastrointestinal	GCB	0	651	
Sample44	T44	Z2114970	Male	63	Nodal	NA	1	1	
Sample45	T45	Z2122930	Female	57	Gastrointestinal	ABC	1	476	
Sample46	T46	Z2129658	Male	64	Gastrointestinal	ABC	0	576	
Sample47	T47	Z2130933	Female	54	Nodal	ABC	0	630	
Sample48	T48	Z2131460	Female	66	Gastrointestinal	ABC	0	651	
Sample49	T49	Z2131475	Female	52	Gastrointestinal	GCB	0	632	
Sample50	T50	Z2131621	Female	55	Nodal	ABC	0	630	
Sample51	T51	Z2137167	Male	71	Gastrointestinal	GCB	0	609	
Sample52	T52	L1900193	Female	68	Nodal	ABC	0	1,424	
Sample53	T53	L1900272	Male	53	Nodal	ABC	1	3,511	
Sample54	T54	L2000538	Male	46	Nodal	ABC	0	821	
Sample55	T55	L2000580	Female	50	Nodal	ABC	0	795	
Sample56	T56	L2000381	Female	76	Nodal	ABC	0	2,044	
Sample57	T57	L2000293	Female	75	Nodal	ABC	0	952	
Sample58	T58	L2000076	Male	50	Nodal	ABC	0	1,074	
Sample59	T59	L2000023	Female	34	Nodal	ABC	0	1,142	
Sample60	T60	L1900498	Female	60	Nodal	ABC	0	1,229	
Sample61	T61	L1900460	Male	69	Nodal	ABC	0	1,253	
Sample62	T62	L1900202	Male	22	Nodal	GCB	0	1,395	
Sample63	T63	L1900008	Female	84	Nodal	ABC	1	3,652	
Sample64	T64	L1900135	Male	64	Nodal	ABC	1	729	
Sample65	T65	L1900138	Female	62	Nodal	ABC	0	1,477	
Sample66	T66	Z2108388	Female	53	Nodal	ABC	0	732	
Sample67	T67	Z2058451	Male	72	Gastrointestinal	GCB	0	820	
Sample68	T68	Z2038723	Male	56	Gastrointestinal	GCB	0	886	
Sample69	T69	Z2054305	Female	54	Nodal	GCB	1	226	
Sample70	T70	Z2017133	Male	39	Gastrointestinal	ABC	0	1,520	
Sample71	T71	Z1953852	Male	30	Gastrointestinal	GCB	1	42	
Sample72	T72	Z1900528	Female	70	Gastrointestinal	ABC	0	1,484	
Sample73	T73	Z1941882	Female	55	Nodal	GCB	0	1,285	
Sample74	T74	Z2165832	Male	77	Gastrointestinal	GCB	0	452	
Sample75	T75	Z2161904	Female	35	Gastrointestinal	ABC	0	498	
Sample76	T76	Q2046427	Male	71	Nodal	GCB	0	780	

Discussion

Based on TCGA DLBCL mutation data, TMB may not independently predict DLBCL prognosis. Previous studies explored the predictive value of a 69-gene panel-TMB, indicating its association with poor prognosis and potential as a predictive indicator when analyzed through a nomogram mode (Chen et al., 2021). Additionally, higher TMB in untreated cancer patients often leads to poor outcomes (Bevins et al., 2020), aligning with our research trend. Although not statistically significant, it is speculated that a small sample size or the need for combined use of TMB with other prognostic factors may yield better predictive effects. Unlike the results of the above two studies, our study screened the genes with differential mRNA expression in the high and low TMB groups through the mutation profile information of DLBC in the TCGA database and screened the predictive genes among them that were highly correlated with prognosis by combining the immune infiltration-related genes. 69-gene panel TMB results (Chen et al., 2021) were obtained by studying the mutation information of the genes, whereas our study mainly screened downstream differentially expressed mRNA genes by high and low mutation genomes, and then got the high and low mutation and prognostic factors affecting DLBCL patients at the transcriptional level.

For the high and low TMB groups, DEGs were primarily enriched in immune cell-mediated immune responses, such as lymphocyte-mediated immunity, immunoglobulin-mediated immune response, B cell-mediated immunity, B cell activation, and immune response activating cell surface receptor signaling pathway. The role of immune cells in DLBCL is complex and multifaceted, involving clinical and histological prognostic factors, immune environment interactions, immune escape strategies, and tumor progression. Previous studies have used computational methods to analyze gene expression profiling (GEP) datasets of 175 DLBCL cases, identifying prognostic genes related to the immune microenvironment, and showed that patients with higher proportions of dendritic cells (DCs) had more prolonged overall survival (OS) (Ciavarella et al., 2019). Our GSEA analysis showed that high TMB groups were mainly enriched in the one-carbon pool by folate (OCPF) and rig I-like receptor (RLR) signaling pathways. Previous study has shown that defects or mutations in genes involved in the OCPF pathway are closely related to DNA methylation and play a crucial role in tumor development (Pan et al., 2021). In recent years, research has explored the correlation between the RLR signaling pathway and tumor mutation burden. Mutations or abnormal expression levels of specific genes in the RLR signaling pathway, such as MAVS, TBK1, and IRF3, can lead to changes in the immune signaling pathway, resulting in a decreased ability of immune cells to recognize and clear malignant tumors (Pecori et al., 2023). The low TMB group is mainly enriched in tight junctions (TJs). Some recent studies have found a specific association between TJs and low tumor mutation burden (Hashimoto & Oshima, 2022).

To elucidate the intrinsic connection between TMB and immune infiltrating cells in the tumor microenvironment, our further studies indicate that the high TMB group has a lower immune score, significant infiltration of M0 and M2 macrophages, activated dendritic cells, and activated immune responses. The above conclusion is consistent with previous literature reporting. Tumor-associated macrophages may provide a growth-promoting microenvironment for malignant B cells and are associated with poor prognosis (McCord et al., 2019). Activated dendritic cells are closely related to higher patient survival rates and prolonged progression-free survival (Ciavarella et al., 2018), which may be because activated dendritic cells can promote T cell activation and enhance tumor-specific cytotoxicity (Garris & Luke, 2020), thereby improving the treatment efficacy for DLBCL patients. However, studies also report a correlation between activated dendritic cells and poorer prognosis (Merdan et al., 2021), which may be because the number of activated dendritic cells is often replaced by other cells in patients with DLBCL, which is also a manifestation of immune system dysregulation in these patients. The low TMB group has an abundance of B cells, follicular helper T cells, regulatory T cells, γδT cells, and various resting immune cells. The study indicated that the impact of inhibiting follicular helper T cells and B cells on tumor immune response is more profound than that of inhibiting CD8+ T cells, indicating that B cells and follicular helper T cells play a critical role in tumor immune response (Hollern et al., 2019). Furthermore, higher levels of γδT cells have been associated with a better prognosis. One study identified two clusters (EC1 and EC2), with EC1 showing an abundance of TP53, MYD88, HIST1H1D, HIST1H1C, KMT2D, and EZH2 mutations, and poor prognosis. EC2 is often accompanied by B2M, CD70, and MEF2B mutations, which are related to DNA damage repair, cytokine-mediated and B cell activation immune signal transduction, elevated CD8+ T cell, γδT cell, and T helper cell levels, increased immune score and immunogenic cell death (ICD) regulatory factors, with good patient prognosis (Wang et al., 2022).

Generally, differences in immunogenicity may lead to differences in immune mechanism activation, and the activation of different types of immune cells in the high TMB group may indicate the inhibition of immune response. It is worth noting that in our study, PIM3 and KMT2D have prominent performances among mutated genes. Previous research has shown that high expression of PIM3 is a poor prognostic factor for DLBCL patients and is associated with common mutated genes in DLBCL, such as MYD88, MYC, and BTK (Wang et al., 2023). KMT2D is a methyltransferase that can reduce the transcriptional activity of specific genes. In a study of DLBCL, somatic mutations in KMT2D were most commonly observed (at 19.5%) and were associated with poor prognosis (Liu et al., 2021).

CD1c has been finally confirmed as a TMB-associated immune gene relevant to the prognosis of DLBCL, and its function has been further explored. CD1c encodes a transmembrane glycoprotein, a member of the CD1 family, and is associated with β2-microglobulin (Moody & Suliman, 2017). Analysis of CD1 expression in B-cell chronic lymphocytic leukemia shows that CD1 mediates immune deficiency, cytokine response polarization, adhesion changes, increased intracellular protein transfer, and leukemia cell processing (Zheng et al., 2002). CD1c+ restricted T cells exhibit potent anti-leukemia activity in mouse models, indicating that this lipid antigen may represent a new target for immunotherapy of hematological malignancies (Lepore et al., 2015). In non-small cell lung cancer (NSCLC), the CD1c+ DCs subset may play an important role in anti-tumor immunity (Lu et al., 2019). A study on early lung adenocarcinoma with EGFR mutation found that the infiltrating T cell types were mainly exhausted and regulatory T cells, which were associated with an increase in dendritic cells expressing the CD1c gene precisely (He et al., 2021). We found one article by searching for CD1b as a potential prognostic biomarker associated with tumor mutation burden and promotes antitumor immunity in lung adenocarcinoma. In this study, the authors showed that CD1b expression is associated with a better prognosis in Lung Adenocarcinoma (LUAD) and promotes anti-tumor immunity by constructing a TMB prognostic model that effectively predicts the prognosis of patients with LUAD, which can be used as a potential prognostic biomarker and immune-related therapeutic target for LUAD (Li et al., 2022). While our study focused on the prognostic value of CD1c in DLBCL, the function of the target gene was verified by various database applications, such as TIMER, CellMiner, konckTF and GTEx. Wet assays confirmed the expression of the target gene at the RNA and protein levels in DLBCL tissue and cell samples. The count of CD1c+ DCs in the blood of gastric cancer patients increases (Liu et al., 2018). In renal cell carcinoma, CD1c+ DCs predict progression-free survival (van Cruijsen et al., 2008). In hepatocellular carcinoma (HCC), the elevation of the ILT4+CD1c+ subset in tumor tissue may play an essential role in immune suppression (Wang et al., 2019). In breast cancer (BRCA), the expression of CD1c correlated with BRCA prognosis and clinical features (Chen et al., 2023). We have identified an increase in CD1c expression in DLBCL tissue, which is associated with poor prognosis according to survival analysis. Although CD1c is a marker of dendritic cell maturation, previous studies have reported CD1c antigen expression in both normal and neoplastic B cells. While CD1c has not been studied as a viable biomarker for DLBCL immunotherapy yet, in our study, we explored the relationship between CD1c expression and prognosis in DLBCL using bioinformatics analysis, revealing that low CD1c expression is indicative of better prognosis. Flow cytometry and immunohistochemistry results demonstrate CD1c antigen expression in a variety of B-cell subgroups, including mature and immature B-cell lines, as well as chronic lymphocytic leukemia (CLL) and hairy cell leukemia (HCL) (Smith, Thomas & Bodmer, 1988). HCL patients with high CD1c expression generally have a poor prognosis, possibly due to HCL cells being over-activated in cases of high CD1c expression, resulting in a more invasive phenotype (Bourguin-Plonquet et al., 2002). Additionally, high CD1c expression in dendritic cells may be closely associated with their immunological function in the immune system and play an essential role in tumor treatment. Therefore, CD1c expression likely has different mechanisms of action regarding prognosis in different cell types, and further research is needed to determine its specific impact.

After conducting further in-depth analysis of CD1c, we found that high amplification of CD1c in B cells and dendritic cells suggests that CD1c mutations inhibit the effective mediation and maintenance of normal immune response by antigen-presenting cells. The poor prognosis of patients with high CD1c expression supports this observation. Targeted gene sensitivity analysis indicates CD1c is associated with clinical sensitivity to various drugs. Molecular research suggests that transcription factors CREB1, AHR, and TOX drive tumor growth and metastasis and are associated with poor prognosis of DLBCL (Huang et al., 2022). The expression of CD1c mRNA and protein levels was verified by wet experiments at cell and tissue levels, indicating that CD1c was more expressed in tumor tissues than in reactive hyperplasia tissues, and the high expression group had a poor prognosis. Studies on the prognostic value of CD1c have shown that high CD1c expression at the mRNA level has a poor prognosis and low CD1c expression has a good prognosis. Prognostic analysis of CD1c in the TIMER database also showed that high mRNA expression within 10 years was associated with poor prognosis. Prognostic analysis of DLBCL clinical samples showed that patients with high CD1c expression at the mRNA level had a poor prognosis, and the same was true for patients with high CD1c expression at the protein level, but the difference was not statistically significant. Based on the experimental results and previous related literature, the possible mechanism and biological function of CD1c in tumor prognosis were explored. Up-regulation of CD1c expression inhibits the effective mediation and maintenance of normal immune responses by antigen-presenting cells, which in turn leads to poor prognosis of patients.

In summary, based on the joint analysis of TMB and immune infiltration, our study has identified immune genes associated with prognosis in DLBCL mutations and explored the inherent correlation between TMB and immune infiltration. CD1c has been identified as a potential biomarker for DLBCL, which may provide new insights for combination therapy.

Conclusions

In conclusion, based on the co-analysis of TMB and immune infiltration, our study identified the immune genes associated with prognosis in DLBCL mutations. It explored the internal correlation between TMB and immune cells infiltrated in the immune microenvironment. CD1c was recognized as a potential marker of DLBCL, which may provide new insights into the immunotherapy of DLBCL. CD1c is also a gene associated with TMB of DLBCL, which predicts poor survival. Bioinformatic analysis shows CD1c is involved in tumor-related signaling pathways and immune and metabolic processes. Thus, the study offers a novel target to investigate the underlying mechanism for diffuse large B cell lymphoma.

Supplemental Information

Supplemental Information 1 Genecloud plot showed mutation information of genes in DLBCL.

Click here for additional data file.

Supplemental Information 2 KEGG pathway analysis revealed that these DEGs were involved in immune-related pathways.

Click here for additional data file.

Supplemental Information 3 Forest map of 13 TMB-related immune genes by univariate Cox analysis.

Click here for additional data file.

Supplemental Information 4 Kaplan Meier survival curves of CD1c expression and prognosis in different datasets.

(A) GSE32918 (P < 0.0001). (B) GSE31312 (P = 0.034). (C) GSE10846 (P = 0.067). (D) GSE53786 (P = 0.041). (E) GSE87371 (P = 0.031). (F) GSE181063 (P =0.00011).

Click here for additional data file.

Supplemental Information 5 Comparison of CD1c mRNA expression between tumor and normal tissues by combined analysis of GETx and TCGA-DLBC database.

Click here for additional data file.

Supplemental Information 6 The differences in clinical characteristics between low- and high-TMB groups were obtained from the TCGA cohort.

Click here for additional data file.

Supplemental Information 7 Top GO items for differentially expressed genes.

Click here for additional data file.

Supplemental Information 8 Top KEGG enrichment pathways for differentially expressed genes ordered by p-value (P < 0.05).

Click here for additional data file.

Supplemental Information 9 CD1c mRNA and protein expression in Diffuse large B cell lymphoma.

Click here for additional data file.

We gratefully acknowledge contributions from the public databases. All the data included in the study was from open databases. We sincerely recognize the TCGA database (https://portal.gdc.cancer.gov/), GEO database (http://www.ncbi.nlm.nih.gov/geo/), ImmPort database (https://www.immport.org/shared/home), UCSC Xena (https://xena.ucsc.edu/), GETx database (https://www.gtexportal.org/home/) and so on for data collection.

Additional Information and Declarations

Competing Interests

Author Contributions

Human Ethics

Data Availability

The authors declare that they have no competing interests.

Xiaoyu Xiang conceived and designed the experiments, performed the experiments, analyzed the data, prepared figures and/or tables, authored or reviewed drafts of the article, and approved the final draft.

Li-Min Gao analyzed the data, authored or reviewed drafts of the article, and approved the final draft.

Yuehua Zhang analyzed the data, authored or reviewed drafts of the article, and approved the final draft.

Qiqi Zhu analyzed the data, authored or reviewed drafts of the article, and approved the final draft.

Sha Zhao analyzed the data, authored or reviewed drafts of the article, and approved the final draft.

Weiping Liu analyzed the data, authored or reviewed drafts of the article, and approved the final draft.

Yunxia Ye analyzed the data, authored or reviewed drafts of the article, and approved the final draft.

Yuan Tang conceived and designed the experiments, analyzed the data, authored or reviewed drafts of the article, and approved the final draft.

Wenyan Zhang conceived and designed the experiments, analyzed the data, prepared figures and/or tables, authored or reviewed drafts of the article, and approved the final draft.

The following information was supplied relating to ethical approvals (i.e., approving body and any reference numbers):

The Ethics Committee on biomedical Research, West China Hospital of Sichuan University approved the study (2020-703).

The following information was supplied regarding data availability:

The data is available at Figshare: Xiang, Xiaoyu (2023). Raw_Data.zip. figshare. Dataset. https://doi.org/10.6084/m9.figshare.22140245.v2.

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
