# Peer review of "Identifying CD1c as a potential biomarker by the comprehensive exploration of tumor mutational burden and immune infiltration in diffuse large B cell lymphoma"

_PeerJ, doi:10.7717/peerj.16618_

## Round 0.1 · original submission · Major Revisions

The manuscript was reviewed by 3 reviewers and all of them very much agreed on the fact that the manuscript is very informative. Two of the reviewers are very much satisfied with the way the manuscript is written, the data presentation, and the interpretations with some limited concerns, however, one of the reviewers asked for a major revision, criticizing the lack of information on the clinical characteristics of the patients (Age, gender, other factors, etc) and low sample size and also the statistical significance in the study. Therefore, I am recommending a major revision of the manuscript. It is important to have the title in the present tense.

·

Basic reporting

in this article, author aimed to identify the potential biomarker using data derived from the different databases, CD1c is the one observed to be different in patients with different tumor mutation burden.
Paper build on the correlation between high and low TMB levels with survival outcomes. However the data didn't reach statistical significance. Although there is a trend. It could be because of low sample size or the difference does not exist due to other variables.
No data on the clinical characteristics of the patients (Age, gender,other factors etc) is provided
Article has lot of spelling mistakes (Backgrouund), grammatic errors and abbreviation are used directly without defining (DLBCL,TME,CNV,Cor etc) it.
Abstract need to be written properly (author wrote what he did in result section of abstract but didn't mention the results of the analysis( line 28,29,31,32).
qPCR was performed on the Normal and DLBCL patients. Comparing the CD1c expression between patients with low and high TMB score can be useful and should be included in data.

Scales in the figure 7 and 8 are not defined makes it harder to understand.

Figure 8 data is not explained well. Author is trying to say that there is positive correlation between drug sensitivity and CD1c expression, and at the same time mention that lower the expression higher the sensitivity. Hence data is not supporting the drug and sensitivity correlation. Positive correlation means higher the expression higher the sensitivity

Experimental design

Sample size to be increased.
qPCR to be done on patients wit low and high TMB score.

Validity of the findings

Results needs to be explained well. Figure 7,8 Y axis should be labelled.

Additional comments

line 243, (rest of 8 genes or 13 genes).

Too many references are cited.

Title should be in present tense.

Reviewer 2 ·

Basic reporting

In this work, Xiang et al. have shown that CD1C is a potential biomarker to analyze the tumour mutational burden in various cancer models. By the combined use of the Immunology Database, Analysis Portal (ImmPort) database and the univariate
Cox analysis from the Gene Expression Omnibus (GEO) database, including three DLBCL datasets, authors screened the various TMB-related immune biomarker. They verified these biomarker functions using various database applications (TIMER, CellMiner, knockTF, GETx).
The manuscript is well-written, engaging, and informative, but it only focuses on computational studies and requires some in-vivo/in-vitro experiments. As a reader, I can understand how CD1C expressions are associated with TMB.
I have some minor comments that need to be addressed by the authors.

1. I am curious to know “Is CD1C is only specific in B-cell lymphoma or the other cancer model. It would be great if the authors could also include my concern in the discussion part.
2. The whole discussion part could be more substantial. Authors need to rewrite.
3. The reference section needs revisions, and the authors must cite more recent papers.
4. Numerous instances in which the spaces between adjacent words are missing. Also, several grammatical things could be improved. I suggest that the whole manuscript be revised thoroughly to improve the manuscript.

Experimental design

Nothing to add

Validity of the findings

nothing to add

·

Basic reporting

Manuscript is well written. Little more information on existing literature will help reader to appreciate the importance of the study. Data presentation, language and interpretations were optimal.

Experimental design

Experimental design is sufficiently explained with proper references. Methods are not described in detail but are crisp to comprehend quickly.

Validity of the findings

Rationale was good. Although statistical significance is shown clinical significance is not that great. But this will add value to the literature on DLBCL and CD1c correlation, which was not reported earliest to the best of my knowledge.

Additional comments

Paper is well organized, graphs are good, reasonable conclusion were drawn.

---

## Round 0.2 · Minor Revisions

I have carefully reviewed your revised manuscript and am pleased to confirm that the authors have diligently addressed most of the reviewers' comments. However, there are a few more concerns related to the current version and authors that need to be addressed to improve the quality of the paper such as:

Previous publications appear to be very similar to the present study. Certainly, when referencing previous studies in a research paper, it's important to clearly establish how the present study differs from them and how it contributes to the existing body of knowledge. This is essential to demonstrate the novelty and significance of your research. Please include in citations, and discuss how the present study is different from the below-mentioned studies and how this study enhances knowledge of CD1c in the prognosis of DLBCL:

Li Z, Feng Y, Li P, Wang S, Liu X, Xia S. CD1B is a Potential Prognostic Biomarker Associated with Tumor Mutation Burden and Promotes Antitumor Immunity in Lung Adenocarcinoma. Int J Gen Med. 2022 Apr 7;15:3809-3826. doi: 10.2147/IJGM.S352851.

Chen C, Liu S, Jiang X, Huang L, Chen F, Wei X, Guo H, Shao Y, Li Y, Li W. Tumor mutation burden estimated by a 69-gene-panel is associated with overall survival in patients with diffuse large B-cell lymphoma. Exp Hematol Oncol. 2021 Mar 15;10(1):20. doi: 10.1186/s40164-021-00215-4."

·

Basic reporting

Results are in better shape now and explained well.

Experimental design

Experimental design looks good

Validity of the findings

Author answered the required question well.

·

Basic reporting

Authors significantly revised the language and basic reporting of data in this revision.

Experimental design

Experiments were sufficiently explained.

Validity of the findings

Data is good statistically scientifically. I am little concerned about clinical validity of this manuscript as K-M curves, I do not see any major difference.

Additional comments

If clinical significance is acceptable, manuscript is good to go.

---

## Round 0.3 · accepted · Accept

Thank you for submitting your work to the journal. The manuscript received positive appreciation from the reviewers and the criticisms raised were adequately addressed in the revised version. Therefore, I am pleased to inform you that I am recommending the acceptance of your manuscript for publication